# OpenReview forum: "RedVisor: Reasoning-Aware Prompt Injection Defense via Zero-Copy KV Cache Reuse"
_ICML.cc/2026/Conference — ICML 2026 regular_

### Official Review · Reviewer_ZHcg · 2026-03-12

**Soundness:** 2
**Presentation:** 3
**Significance:** 2
**Originality:** 2
**Overall Recommendation:** 4
**Confidence:** 4

**Summary:**

This paper proposes RedVisor, a two-phase defense framework against prompt injection (PI) attacks on LLMs. A lightweight gated attention adapter is placed exclusively at the top transformer layer of a frozen LLM backbone. In Phase 1 (Inspection), the adapter is activated to generate an explainable reasoning trace that localizes and articulates the injection. In Phase 2 (Generation), the adapter gate is set to zero, reverting the model to its original pretrained state; the model then generates a response conditioned on the Phase 1 reasoning as an "in-context guardrail." Because the adapter resides only at the final layer, the backbone's KV cache from Phase 1 is directly reusable in Phase 2 without recomputation. The paper further integrates this mechanism into the vLLM serving engine with custom kernels. Experiments on Alpaca-Farm, AgentDojo, and NQ-simplified across three 7-8B backbones (Llama-3, Mistral, Qwen-2.5) show competitive detection accuracy, low attack success rates, preserved utility on AlpacaEval 2.0, and >2× latency improvement over decoupled detection pipelines.

**Compliance With Llm Reviewing Policy:**

Affirmed.

**Final Justification:**

The follow-up experiments addressed my concerns. I think this paper can be accepted.

**Key Questions For Authors:**

check the weakness

**Limitations:**

Partially addressed. The Impact Statement acknowledges that "no defense is impenetrable" and discusses over-reliance risk. However, several critical limitations are not discussed: (1) the reliance on template-based training and its implications for OOD generalization; (2) the absence of false positive analysis; (3) the potential for security reasoning tokens to interfere with complex downstream tasks; (4) the lack of comparison to direct CoT fine-tuning, which is arguably the simplest and most natural alternative; and (5) the complete omission of reasoning models from the discussion, despite their direct relevance to security-aware chain-of-thought.

**Strengths And Weaknesses:**

### Strengths

**S1. Compelling preliminary insight.** The observation in Table 1 that providing ground-truth security reasoning reduces ASR to 0% across all models and all five attack categories is a clean and actionable finding. It sharply isolates the bottleneck: LLMs can already refuse injections—they simply fail to spontaneously detect them. This motivates the entire approach well.

**S2. Elegant KV cache reuse enabled by top-layer-only placement.** Placing the adapter exclusively at the final layer is the key design insight. It simultaneously enables (a) zero-copy KV cache reuse across phases, since layers 1 through L−1 are untouched, and (b) exact restoration of the original model when the gate is zeroed. The formal latency analysis (Equations 11–12) confirms only a ~1/L overhead for the second phase's prefill, which is negligible for deep models.

**S3. Solid systems contribution.** The vLLM integration—with vectorized masking for topological invariance, tail-anchored pattern matching for phase detection, and atomic phase coupling—is a meaningful engineering effort that bridges concept and deployment.

**S4. Broad experimental coverage.** Three benchmarks spanning instruction following, agentic tool use, and RAG; three backbone models; six baselines from both detection and prevention paradigms; five attack types plus adaptive GCG—this is comprehensive within the tested scope.

**S5. Informative ablation.** The ablation (Table 9) revealing that even the "None" baseline (raw backbone with the reasoning template but no adapter) significantly reduces ASR is a particularly honest and revealing result. It suggests much of the defense value comes from the reasoning-before-answering paradigm itself, with the adapter refining it further.

---

### Weaknesses

**W1 (Major). The most natural and strongest baseline is missing: directly fine-tuning the backbone's own CoT to include security reasoning.** The paper's entire motivation rests on the "alignment tax" of fine-tuning, yet it never quantifies this tax for the most directly comparable approach. The paper shows that ground-truth reasoning yields 0% ASR (Table 1), and acknowledges that even a reasoning template without any adapter significantly reduces ASR (Table 9, "None" row). The logical next step is to ask: **what if we simply LoRA-fine-tune the backbone itself on the same security-reasoning training data?** This would produce a single model that natively generates security analysis in its CoT before responding—no adapter, no phase switching, no custom kernels, no KV cache reuse needed.

The paper argues that fine-tuning degrades utility, pointing to StruQ (which drops to 56.75% WinRate on Qwen). But StruQ uses a fundamentally different training paradigm (injection-augmented instruction tuning with structured delimiters), not reasoning-augmented CoT training. The comparison is apples-to-oranges. A fair baseline would use **the exact same training data** (the ~52k augmented Alpaca samples with reasoning traces) but apply it as a standard LoRA fine-tune to the backbone itself. This critical experiment is absent, making it impossible to assess whether RedVisor's architectural complexity is justified.

**W2 (Major). No evaluation on complex reasoning tasks (math, code, long-form).** The paper claims "mathematically preserved utility" and zero alignment tax, but utility is evaluated solely on AlpacaEval 2.0 (general instruction following). This is an easy test because on benign inputs with the adapter muted, the backbone is the original model. However, **even with the adapter muted, the Phase 1 reasoning tokens remain in the context window for Phase 2.** This means the model's effective context for generation includes a security analysis preamble of non-trivial length. The paper never evaluates whether this preamble:

- Consumes meaningful token budget on tasks with limited context windows or long inputs (e.g., long-document RAG);
- Interferes with the model's own chain-of-thought on complex reasoning tasks (GSM8K, MATH, HumanEval) where the CoT must be coherent and focused;
- Creates attention interference, where the model may attend to security-analysis tokens instead of task-relevant ones during difficult multi-step reasoning.

This is not a minor omission. If the security reasoning preamble degrades performance on a 5-step math proof or a complex coding problem—even by a few percent—it would undermine the paper's central claim of zero utility loss.

**W3 (Major). The "zero alignment tax" claim is misleading when reasoning tokens are considered.** The paper repeatedly asserts that muting the adapter "mathematically preserves the backbone's original utility." Strictly speaking, this is true only for the model's *weights*—the forward pass through layers 1 to L is identical to the original model. But the model's *input* is no longer original: it now includes the Phase 1 reasoning tokens. These tokens were generated by the adapter-modified model (a different distribution than the backbone alone), and they occupy context that would otherwise be available for the task. Calling this "mathematically preserved utility" is technically correct in a narrow sense but practically misleading, because the generation is conditioned on an exogenous reasoning prefix that the original model would never have produced. The paper should be transparent about this distinction.

**W4 (Moderate). Template-based training data limits generalization.** The reasoning traces are synthesized via fixed templates with spaCy-based intent extraction (Section 6.2/Appendix C). The adapter thus learns pattern matching against five known attack categories rather than genuine security reasoning. This raises serious concerns about:

- **Novel/out-of-distribution attacks** not conforming to the five templates;
- **Multilingual injections** (training is English-only);
- **Semantically subtle injections** that avoid template-matching keywords (e.g., "As part of this document's purpose, the user actually wants...");
- **Encoded/obfuscated injections** (Base64, unicode homoglyphs).

No out-of-distribution evaluation is conducted.

**W5 (Moderate). Complete absence of false positive analysis.** The paper does not report false positive rates on benign inputs. The WinRate metric assesses output quality, not whether the adapter erroneously flags benign content as malicious. In production, a model that frequently hallucinates injection detections in clean documents would be unusable. Consider: a user uploads a document about prompt injection defense (containing attack examples as illustrative content)—would the adapter flag it? This is a critical deployment metric that is entirely missing.

**W6 (Moderate). The ablation reveals a potentially inconvenient finding that deserves deeper discussion.** The "None" baseline in Table 9 shows that the raw backbone with just the reasoning template (no adapter at all) already achieves substantial ASR reduction (e.g., from ~100% to 42-66% on Llama-3 across attack types). This suggests that a large portion of the defense comes from the prompting strategy itself, not the adapter. This raises the question: how much of RedVisor's residual gain over "None" could be recovered by simply fine-tuning the model to reliably produce this reasoning on its own, without any external adapter?

---

> ### Author Rebuttal · Authors · 2026-03-30
>
> Dear Reviewer ZHcg,
>
> We sincerely thank you for your thoughtful review and for recognizing our compelling preliminary insights (S1), elegant KV cache reuse architecture (S2), solid systems contribution (S3), and broad experimental coverage (S4). We address your insightful critiques below.
>
> **1. Direct CoT Fine-Tuning Baseline (W1, Limitations 4 & 5)**
> We conducted the exact experiment of directly LoRA-fine-tuning the backbone on the security-reasoning data. While the fine-tuned backbone successfully learns to generate reasoning traces, it suffers from severe **mode collapse** during the subsequent generation phase. Because the model is forced to internalize both security reasoning and general task execution within the same frozen weights, it often fails to transition back to the user's actual task, instead continuing to output security illustrations. This causes a catastrophic drop in utility, significantly worse than standard prevention baselines (StruQ/SecAlign).
>
> **AlpacaEval 2.0 WinRate:**
> | Backbone LLM | Raw | StruQ | SecAlign | RedVisor | LoRA (Direct CoT) |
> | :-: | :-: | :-: | :-: | :-: | :-: |
> | Llama3-8B | 87.1% | 69.8% | 79.3% | 86.4% | **53.9%** |
> | Mistral-7B | 86.4% | 70.2% | 71.5% | 83.6% | **49.0%** |
> | Qwen-2.5-7B | 91.2% | 57.1% | 67.1% | 85.4% | **58.7%** |
>
> This definitively proves that RedVisor's architectural design, specifically the physically mutable adapter, is strictly necessary to decouple detection from generation and preserve utility.
>
> **2. Evaluation on Complex Reasoning Tasks (W2, Limitation 3)**
> Regarding complex reasoning tasks, we emphasize that our evaluation already covers the most realistic and standard complex scenarios in the prompt injection community, specifically long-document RAG and Agentic workflows. However, to further strengthen our claims and directly address your concern regarding Chain-of-Thought (CoT) interference, we conducted an additional evaluation on the **GSM8K** dataset (1.3k test set).
> On benign queries, RedVisor’s Phase 1 trace is microscopic (e.g., *"No injection detected"*). As shown below, this brief preamble does not meaningfully consume token budget or derail the backbone's native CoT, preserving complex math reasoning far better than fine-tuning alternatives:
>
> **GSM8K Exact Match Accuracy:**
> | Backbone LLM | Raw | StruQ | SecAlign | RedVisor |
> | :-: | :-: | :-: |:-: | :-: |
> | Llama3-8B | 80.3% | 74.4% | 75.8% | **79.2%** |
> | Mistral-7B | 40.6% | 38.2% | 38.5% | **38.9%** |
> | Qwen-2.5-7B | 83.1% | 75.1% | 77.1% | **81.8%** |
>
> **3. Utility and Alignment Claims (W3)**
> Regarding the "zero alignment tax" claim, we clarify that this refers specifically to the strict preservation of the backbone's model parameters. While the Phase 1 reasoning trace does slightly append the input context for Phase 2, our empirical results across multiple benchmarks demonstrate that this brief preamble has a negligible impact, and the model's inherent utility is largely maintained. To avoid any potential ambiguity, we will refine our phrasing in the revision to explicitly state "parameter-level preservation".
>
> **4. Template-Based Training & Generalization (W4, Limitation 1)**
> We clarify that our training data is not rigidly templated. While we follow the standard 5 attack categories of the prompt injection community, the *injections themselves* are diverse, human-written, real-world instructions. For example, our "ignore" attacks utilize a wide variety of semantic prefixes (e.g., *"Oh, wait, I just got an urgent message..."*) rather than a single fixed string. Furthermore, AgentDojo tests open-ended tool-use completely unseen during training. Finally, we evaluated RedVisor against entirely new OOD attacks (Multi-Lingual, Base64/Unicode Obfuscation, and XML-tag mimicry) and observed no performance drop (0.00 ASR). *(Please see our rebuttal to Reviewer USTm's W1 for the full OOD table).*
>
> **5. False Positive Analysis (W5, Limitation 2)**
> RedVisor maintains a low False Positive (FP) rate on clean queries (e.g., ~3-6% across benchmarks; detailed in our rebuttal to Reviewer mi4p's W4). Crucially, a Phase 1 false positive does **not** render the system unusable. Unlike standard filters that trigger a hard refusal (e.g., "I cannot answer this"), RedVisor simply conditions the Phase 2 backbone to ignore the falsely flagged segment. The model still successfully executes the core user query using the remaining benign context, ensuring robust production utility.
>
> **6. The "None" Baseline Ablation (W6)**
> While Table 9 shows the reasoning prompt alone provides *some* defense, a drop from ~100% to ~42% ASR is entirely unacceptable for a security system. The raw backbone fails as a reliable detector (e.g., achieving only 0.14 Recall on Naive attacks). The adapter is absolutely mandatory to bridge this gap, driving the detection accuracy up and crushing the residual ASR to near-zero (0.00), achieving the "last mile" of critical security that prompting alone cannot provide.

---

> > ### Author Rebuttal · Reviewer_ZHcg · 2026-04-02
> >
> > The reported "mode collapse" — where the LoRA model keeps outputting security analysis instead of answering the user — is a textbook symptom of training data imbalance, not an inherent limitation of direct CoT fine-tuning. We strongly suspect the model was fine-tuned on security-reasoning-only targets without (1) mixing in general instruction-following data (e.g., HelpSteer, SafeRLHF) to prevent catastrophic forgetting, or (2) using complete training samples (<think>analysis</think> + normal response) so the model learns the full reasoning-then-answering pipeline. No hyperparameters are reported either. This contradicts the basic practices of modern safety alignment — virtually all SOTA models (GPT-4, Llama-3-Instruct, DeepSeek-R1) incorporate safety reasoning during post-training without such collapse. A single under-specified failed experiment cannot support the strong claim that the adapter is "strictly necessary."
> >
> > The authors should not focus on the results now. Show me your method's benefit compared to in-thinking security analysis

---

> > > ### Author Response · Authors · 2026-04-04
> > >
> > > We sincerely thank the reviewer for their rigorous engagement regarding the direct in-thinking (CoT) fine-tuning baseline. To address your concerns, we want to clarify the fundamental motivation behind RedVisor and highlight the structural limitations of direct LoRA fine-tuning for this specific task.
> > >
> > > As demonstrated in our preliminary findings (Table 1), if ground-truth security reasoning is appended to the context of a *raw, unaltered* backbone, the LLM perfectly neutralizes injections (0% ASR) while maintaining its innate, high general utility. However, the raw LLM's zero-shot security reasoning ability is too weak to generate these traces reliably. Therefore, we must strengthen this capability. Two practical methods exist: (1) direct LoRA fine-tuning, and (2) a decoupled adapter (our proposed RedVisor).
> > >
> > > While LoRA is a standard approach, it exhibits critical limitations for prompt injection defense:
> > >
> > > **1. Inevitable Utility Drop Despite Extensive Data Mixing**
> > >
> > > We acknowledge that our previous LoRA baseline was trained solely on reasoning data. To rigorously test if the observed mode collapse was merely a data-mixing issue, we conducted a new LoRA baseline experiment heavily optimized for general utility.
> > >
> > > We applied LoRA to all linear layers with rank $r=32$ (yielding ~80M trainable parameters, deliberately matching RedVisor's capacity).
> > > Crucially, we post-trained this baseline on a comprehensive mixture of our security reasoning dataset *and* the full **Alpaca-Cleaned** instructional dataset. To explicitly teach the model the complete pipeline and prevent catastrophic forgetting, the training samples were formatted as concatenated sequences (i.e., `[security reasoning trace] + [ground-truth answer]`).
> > >
> > > Despite this extensive data mixing and ample parameter capacity, the direct CoT baseline still suffers a severe utility drop on benign instructions:
> > >
> > > **AlpacaEval 2.0 WinRate:**
> > > | Backbone LLM | Raw (Phase 2 state) | RedVisor | LoRA (Security + Alpaca-Cleaned) |
> > > | :--- | :--- | :--- | :--- |
> > > | Llama-3-8B | 87.1% | 86.4% | **72.4%** |
> > > | Mistral-7B | 86.4% | 83.6% | **64.5%** |
> > > | Qwen-2.5-7B | 91.2% | 85.4% | **75.8%** |
> > >
> > > *Analysis:* Forcing the model to continuously manage both strict, structured security protocols and open-ended generation within the same continuous weights fundamentally disrupts its original instruction-following distribution.
> > >
> > > **2. Data Availability and Training Overhead**
> > >
> > > To even attempt to salvage the utility of a LoRA-fine-tuned model, developers must possess and curate massive instructional datasets containing **high-quality ground-truth answers** (like Alpaca-Cleaned) to mix with the security data. In real-world deployments, such high-quality alignment data might be proprietary, domain-specific, or entirely unavailable to the end-user. Furthermore, forcing the model to continuously "retrain" on this massive mixed dataset dramatically inflates training time, GPU requirements, and memory overhead.
> > >
> > > Conversely, RedVisor's decoupled architecture maximizes training efficiency. Because Phase 2 generation operates strictly on the frozen, unadulterated base model, RedVisor *only* needs to be trained on the constructed security reasoning dataset (Section 6). This completely eliminates the computational overhead of retraining on original instructional answers and structurally guarantees that the model avoids utility degradation.
> > >
> > >
> > > **3. Localized Isolation vs. Deep Entanglement**
> > >
> > > RedVisor also offers distinct structural benefits. LoRA distributes weight updates deeply across all representation layers, whereas RedVisor quarantines security parameters exclusively at the topmost layer.
> > >
> > > * **More Convenient Updating & "Unlearning":** Retraining modules for new attacks is more straightforward with RedVisor. While updating a distributed LoRA risks unintended interference with core representations, RedVisor strictly modifies the final decision boundary. With deeper layers frozen, it is safer and easier to update or "unlearn" data without risking cascading shifts.
> > > * **Easier Interpretability and Control:** Concentrating security logic at a final-layer bottleneck makes it far easier to trace activations and red-team decision boundaries. Conversely, analyzing a LoRA defense is more complex, requiring the disentanglement of weight shifts across all hidden layers.
> > >
> > >
> > > **Conclusion & Future Work**
> > >
> > > We acknowledge that natively integrating CoT-based safety reasoning is a highly promising future direction for massive-scale models (e.g., DeepSeek-R1, GPT-4) equipped with extensive post-training datasets. However, for smaller-scale open-weight LLMs with limited deployment data, RedVisor provides a mathematically cleaner, highly efficient, and vastly more practical solution. We will include these new LoRA comparisons and discussions in the revised Appendix.

---

### Official Review · Reviewer_mi4p · 2026-03-12

**Soundness:** 3
**Presentation:** 3
**Significance:** 3
**Originality:** 3
**Overall Recommendation:** 4
**Confidence:** 3

**Summary:**

This paper proposes RedVisor, a defense framework for prompt injection. The method separates security analysis from final response generation into two stages: in Stage 1, a lightweight adapter placed on top of a frozen backbone produces interpretable localization and reasoning; in Stage 2, the adapter is muted and the original backbone generates the final response conditioned on the Stage-1 reasoning trace. The paper argues that this design balances interpretability, utility, and efficiency, while reusing the KV cache to reduce two-stage inference overhead. The reported empirical results are generally strong across multiple backbones and benchmarks.

**Compliance With Llm Reviewing Policy:**

Affirmed.

**Final Justification:**

I think the authors’ rebuttal addresses my main concern. I would keep my original score: weak accept.

**Key Questions For Authors:**

1. Please clarify the exact meaning of “mathematically preserves the backbone’s original utility.” Is the claim about parameter-level non-destructiveness, or strict behavioral equivalence?
2. Please provide stronger evaluation on more open-ended, non-templated, cross-domain, or human-written attacks.
3. Please clarify whether the latency comparisons were conducted under equally optimized implementations, batching, concurrency, and caching conditions.
4. Please report false positive rates, over-refusal rates, and performance on clean long-context settings.
5. Please provide more detail on the white-box adaptive attack setup, including the optimization target, attack budget, and success criterion.

**Limitations:**

The paper does note that the method should not be treated as a silver bullet, but the limitations discussion is still incomplete. It would benefit from a more explicit discussion of the synthetic-to-real gap, false positives and over-defensiveness, the extra cost of reasoning tokens, and limitations in multilingual and tool-use settings.

**Strengths And Weaknesses:**

Strengths.
A key strength of this paper is the tight coupling between model design and serving efficiency. The top-layer adapter plus muted second stage is a sensible and somewhat novel design, especially because it enables cache reuse. The experimental scope is reasonably broad, covering detection, ASR, utility, latency, and ablations. Empirically, RedVisor appears effective across multiple attack types while being more efficient than a decoupled two-stage version, which makes the work practically relevant.

Weaknesses.
My main concern is that several central claims are stronger than the current evidence. First, the statement that the method “mathematically preserves the backbone’s original utility” is too strong: Stage 2 conditions on additional reasoning and a transition instruction, so the behavior is not identical to that of the raw backbone. Second, the latency discussion is useful for intuition, but the “theoretical lower bound” wording still seems overstated without stricter and fully controlled system comparisons. Third, both the attacks and the reasoning supervision are heavily templated, so stronger evidence is needed for generalization to more realistic and open-ended prompt injection settings. Finally, the current utility evaluation does not fully characterize false positives, over-defensiveness, or long-context/agentic behavior.
Overall, I find the paper promising and practically meaningful, but the strongest claims should be stated more carefully.

---

> ### Author Rebuttal · Authors · 2026-03-30
>
> Dear Reviewer mi4p,
>
> We sincerely thank you for your thoughtful review and for highlighting that RedVisor's tight coupling of model design and serving efficiency is a sensible and novel approach that makes the work practically relevant. Below, we address your specific questions.
>
> **1. Utility Preservation Claims (W1, Q1)**
> Regarding the phrase "mathematically preserves the backbone's original utility", we clarify this refers strictly to **parameter-level non-destructiveness**. By explicitly muting the adapter ($\alpha=0$) in Phase 2, the backbone's original weights remain unaltered, avoiding the structural degradation inherent to fine-tuning (e.g., SecAlign). We will revise this phrasing in the manuscript to prevent any ambiguity regarding strict behavioral equivalence.
>
> **2. Latency and System Comparisons (W2, Q3)**
> We confirm all methods were evaluated under equally optimized conditions, natively integrated into vLLM with automatic continuous batching and KV caching. Thus, the empirical latency gains (Figure 4) are fair and directly stem from RedVisor's architecture. As formalized in **Section 5.3 (Theoretical Complexity)**, RedVisor fundamentally eliminates the **"double prefill" penalty ($2 \cdot \mathcal{P}(L)$)** and **inter-device communication costs ($\mathcal{T}_{comm}$)** strictly required by decoupled baselines. We will temper the "theoretical lower bound" wording to emphasize these quantifiable system-level efficiencies. *(Please see our rebuttal to Reviewer USTm's W2 for absolute overhead details compared to an undefended backbone).*
>
> **3. Attack Generalization and Templates (W3, Q2)**
> We clarify that our training and evaluation do not rely on rigid templates. We follow the well-established taxonomy of prior works (e.g., StruQ), but the instances within these categories are highly diverse. For example, our **"naive" attacks** consist of flexible, human-written malicious instructions organically inserted into contexts. Similarly, our **"ignore" attacks** utilize a wide variety of diverse semantic prefixes, rather than relying on a single, rigid string like *"Ignore previous instructions"*. Furthermore, AgentDojo tests open-ended, agentic **tool-use** scenarios completely unseen during training. To further prove robustness, we also evaluated RedVisor against OOD Multi-lingual, Obfuscated, and XML-tag mimicry injections, observing no performance drop (*detailed in our rebuttal to Reviewer USTm's W1*).
>
> **4. False Positives and Clean Long-Context Utility (W4, Q4)**
> To evaluate over-defensiveness, we measured the False Positive (FP) rate by treating RedVisor as a binary detector on clean queries (flagged as an FP if it fails to output "No injection detected"):
>
> | Backbone LLM | Alpaca-Farm | NQ | AgentDojo |
> | :-: | :-: | :-: | :-: |
> | Llama3-8B | 3.4% | 7.8% | 1.9% |
> | Mistral-7B | 6.3% | 9.2% | 3.1% |
> | Qwen-2.5-7B | 4.3% | 8.6% | 2.7% |
>
> To address utility in clean, long-context/RAG settings, we evaluated performance on Natural Questions (NQ). We adapted the AlpacaEval framework by replacing standard instructions with the NQ questions, and used GPT-4.1-mini to generate the baseline reference answers for our WinRate comparison:
>
> | Backbone LLM | Raw | StruQ | SecAlign | RedVisor |
> | :-: | :-: | :-: | :-: | :-: |
> | Llama3-8B | 47.2% | 31.0% | 33.8% | 42.1% |
> | Mistral-7B | 30.8% | 19.3% | 21.4% | 26.7% |
> | Qwen-2.5-7B | 51.2% | 34.6% | 38.0% | 45.6% |
>
> RedVisor significantly outperforms prevention baselines, closely trailing the undefended backbone. Importantly, Phase 1 false positives minimally impact Phase 2 utility: rather than triggering a hard over-refusal, RedVisor simply conditions the backbone to ignore the falsely flagged segment, successfully preserving its ability to answer the core query using the remaining context.
>
> **5. White-Box Adaptive Attack Setup (Q5)**
> To rigorously evaluate under a worst-case white-box threat model, we tailored the GCG attack to independently stress-test both phases:
> * **Targets:** **1) GCG$_{LLM}$:** Optimizes an adversarial suffix on the target backbone, minimizing the negative log-likelihood (NLL) of generating the malicious response (testing Phase 2 resilience). **2) GCG$_{classifier}$:** Optimizes a separate suffix on the adapter, minimizing the NLL of outputting exactly "No injection detected" (testing Phase 1 vulnerability).
> * **Budget:** We align with standard configurations: 500 optimization iterations, evaluating a batch size of 512 candidate replacements from the top-$k=256$ token gradients, with a 20-token suffix constraint.
> * **Criterion:** Evaluated strictly. For *GCG$_{LLM}$*, Attack Success Rate (ASR) measures if the target string is elicited. For *GCG$_{classifier}$*, we measure the rate the adapter is forced into a "clean" classification despite an injection. RedVisor's near-zero ASR across both independent attacks demonstrates the structural robustness of the isolated reasoning phase.

---

> > ### Author Rebuttal · Reviewer_mi4p · 2026-04-02
> >
> > Thank you for your detailed response and patient clarification.
> > I find your rebuttal satisfactory, so I will maintain my original score.

---

### Official Review · Reviewer_USTm · 2026-03-13

**Soundness:** 3
**Presentation:** 3
**Significance:** 3
**Originality:** 3
**Overall Recommendation:** 4
**Confidence:** 4

**Summary:**

This paper proposes RedVisor, a two-phase defense framework against prompt injection (PI) attacks on LLMs. The key idea is to equip a frozen LLM backbone with a lightweight, removable gated self-attention adapter placed only at the top transformer layer. In Phase 1 (Inspection), the adapter is active and generates an explicit reasoning trace that localizes and explains any detected injections. In Phase 2 (Generation), the adapter is muted (via a binary mask), reverting the model to its original pre-trained state, and the response is generated conditioned on the reasoning from Phase 1. Because the adapter sits only at the top layer and is toggled via a mask rather than removed structurally, the KV cache from Phase 1 can be directly reused in Phase 2 without re-computation ("zero-copy KV cache reuse"). The authors integrate this into the vLLM serving engine and evaluate on three benchmarks (Alpaca-Farm, AgentDojo, NQ-simplified) across three 7-8B parameter backbones (Llama-3-8B, Mistral-7B, Qwen-2.5-7B), showing strong detection accuracy, low attack success rates, preserved utility, and improved latency compared to decoupled detection-then-generation baselines.

**Compliance With Llm Reviewing Policy:**

Affirmed.

**Final Justification:**

My concerns have been adequately addressed, thus I have decided to maintain my positive recommendation.

**Key Questions For Authors:**

1. What is the absolute latency overhead of RedVisor compared to a completely undefended vanilla backbone? Figure 4 only compares against other defense baselines, making it hard to assess the true cost of the reasoning phase for end users.
2. How does RedVisor perform on injection strategies that are qualitatively different from the five training categories? For example, injections that use subtle semantic manipulation (e.g., biasing the model's response without explicit imperative commands), multi-lingual injections, or injections that mimic the XML-tag structure used by RedVisor itself to confuse the boundary parsing?
3. (Suggestion) The paper claims a white-box threat model, but the adaptive attack evaluation is limited to GCG with two variants. It would strengthen the paper to evaluate against more sophisticated adaptive attacks that specifically target the two-phase architecture—for instance, attacks that craft injections designed to produce misleading reasoning traces (e.g., the reasoning says "No injection detected" but the injected content still influences Phase 2 generation through the cached KV states), or attacks that exploit the fixed transition instruction (Itrans) as a known anchor point.

**Limitations:**

Yes. The authors acknowledge that "no defense is impenetrable" and warn against over-reliance. However, the limitations discussion could be more specific about known failure modes. The paper does not adequately discuss: (1) the overhead of the reasoning phase on user-perceived latency; (2) the risk that the fixed reasoning templates create predictable patterns an adaptive attacker could exploit; and (3) generalization to injection types beyond the five training categories.

**Strengths And Weaknesses:**

Strengths:
- S1: The core architectural insight is clean and well-motivated. Placing the adapter exclusively at the top layer is a principled design choice that simultaneously enables KV cache reuse (since lower-layer key/value states are identical across phases) and guarantees mathematical preservation of the backbone's utility when the adapter is muted. This is a genuine contribution over standard LoRA/adapter approaches that modify parameters at every layer.
- S2: The paper provides a convincing preliminary study (Table 1) showing that when ground-truth reasoning about injections is appended to the context, standard models achieve 0% ASR across all attack types. This strongly motivates the "reasoning-before-answering" paradigm and anchors the entire approach in an empirical observation rather than pure intuition.
- S3: The systems-level contribution is substantial. The integration into vLLM with vectorized masking (to preserve CUDA graph compatibility), tail-anchored pattern matching on GPU tensors, and atomic phase coupling in the scheduler are practical engineering contributions that make the approach deployable. The theoretical latency analysis (eliminating the double-prefill cost) is clearly formulated.
- S4: The experimental evaluation is comprehensive, covering three diverse benchmarks (instruction following, agentic workflows, RAG), three backbone models, six baselines spanning both detection-based and prevention-based paradigms, and both rule-based and adaptive (GCG) attacks. The results consistently favor RedVisor across detection accuracy, ASR, utility preservation, and latency.

Weaknesses:
- W1: The training data and test data share concerning overlap. RedVisor is trained on the Alpaca-Cleaned dataset with synthetic reasoning traces, and then evaluated on Alpaca-Farm (a subset of 208 samples from the same Alpaca distribution). While the paper mentions "zero data leakage" between train and test, the attack patterns used during evaluation (Naive, Ignore, Escape, Completion, Multi-round) are the same five categories used during training. This raises the question of whether RedVisor would generalize to genuinely novel, out-of-distribution injection strategies that do not fall into these predefined categories. The AgentDojo evaluation partially addresses this (using "Important Instructions" attack), but a more systematic evaluation on truly unseen attack types would strengthen the claims.
- W2: The reasoning trace generation adds non-trivial latency that is somewhat downplayed. The paper's latency comparison (Figure 4) shows RedVisor is faster than decoupled baselines, but this comparison is somewhat favorable by design: the baselines require double prefill on separate GPUs, while RedVisor avoids this. A fairer comparison would also show the absolute overhead of RedVisor (Phase 1 reasoning + Phase 2 generation) versus a vanilla backbone with no defense. The reasoning traces can be lengthy (per the templates in the appendix), and generating them autoregressively adds to time-to-first-useful-token for the end user.

---

> ### Author Rebuttal · Authors · 2026-03-30
>
> Dear Reviewer USTm,
>
> We sincerely thank you for your constructive review and for recognizing RedVisor's principled zero-copy architecture (S1), empirical motivation (S2), practical vLLM integration (S3), and comprehensive evaluation (S4). Below, we address your insightful questions regarding generalization, absolute latency, and adaptive attacks.
>
> **1. Generalization and Unseen Attack Types (W1, Q2, Limitations)**
> Regarding the dataset, we strictly follow the established evaluation protocol of prior works like StruQ and SecAlign, utilizing Alpaca-Cleaned for training and Alpaca-Farm for testing. We emphasize that while similarly named, these are distinct datasets with no data leakage.
>
> Regarding attack diversity, the five categories evaluated represent the standard attack taxonomy in current literature. However, we agree that testing on out-of-distribution (OOD) attacks strengthens the paper. Per your suggestion, we evaluated RedVisor against new OOD strategies on Alpaca-Farm. As shown in the table below, RedVisor generalizes exceptionally well:
>
> | Backbone LLM | Multi-Lingual (RL / ES / ASR) | Obfuscated (RL / ES / ASR) | XML-Tag (RL / ES / ASR) |
> | :--- | :--- | :--- | :--- |
> | Llama3-8B | 0.99 / 0.99 / 0.00 | 0.93 / 0.92 / 0.00 | 0.99 / 0.98 / 0.00 |
> | Mistral-7B | 0.97 / 0.98 / 0.00 | 0.91 / 0.90 / 0.00 | 0.97 / 0.97 / 0.00 |
> | Qwen-2.5-7B | 0.98 / 0.99 / 0.00 | 0.93 / 0.93 / 0.00 | 0.97 / 0.97 / 0.00 |
>
> * **Multi-Lingual Injections:** We randomly translated the injections into Chinese and Japanese and observed no performance drop. These injections are often easier for RedVisor to flag due to the stark semantic discrepancy between the benign query and the malicious multi-lingual context.
> * **Obfuscated Injections:** We replaced the original injections with Base64 or Unicode encodings. We again observed 0.00 ASR. While detection accuracy (RL/ES) drops slightly, the threat is naturally mitigated because the backbone LLMs struggle to natively understand and execute the heavily encoded semantics.
> * **XML-Tag Mimicry (Limitation 2):** Attackers attempting to confuse boundary parsing by mimicking our `<user_query>` or `[L1]` tags completely fail (0.00 ASR). Because our preprocessing pipeline strictly relies on the NLTK tokenizer to assign segment IDs *after* the raw text is ingested, injected tags are merely treated as standard text content within a designated segment, structurally preventing bypass.
> * **AgentDojo:** Following the PromptLocate baseline, we emphasize that the evaluated **"Important Instructions" attack already represents the highest success rate in that benchmark**; RedVisor's robust defense against this confirms its strength against lesser attack variants.
>
> **2. Absolute Latency Overhead vs. Vanilla Backbone (W2, Q1, Limitations)**
> To clarify the absolute latency compared to a vanilla, undefended backbone, RedVisor introduces a highly controlled and bounded overhead to the user's time-to-first-useful-token. Because RedVisor's zero-copy architecture completely avoids the redundant prompt-recomputation costs of standard baselines, the discrepancy with a raw backbone stems entirely from two specific areas:
>
> * **Parameter and Compute Overhead:** Our adapter introduces only ~70M parameters to an 8B backbone (a <1% increase). Consequently, the extra computational cost (FLOPs) during the initial prompt processing is negligible and does not meaningfully bottleneck the system's throughput.
> * **Overall User-Perceived Latency:** The primary source of extra latency is the autoregressive generation of the internal reasoning trace. Based on our environment (2x NVIDIA A100-40G GPUs via vLLM), decoding this trace delays the first user-visible token by only ~0.6 to 0.8 seconds for standard localized attacks (e.g., naive), and ~2.5 seconds for complex, multi-segment attacks (e.g., multi-round completion).
>
> Given that RedVisor reduces the Attack Success Rate (ASR) from 40-60% (on undefended or fine-tuned baselines) to near-zero (0.00) against adaptive white-box attacks, this absolute latency increase is a highly practical and acceptable trade-off for critical security.
>
> **3. Sophisticated Adaptive Attacks (Q3)**
> To address your suggestion of evaluating attacks targeting our two-phase architecture (e.g., exploiting cached KV states or the $I_{trans}$ instruction), we evaluated two advanced attacks: **AutoDAN-PI** (White-Box) and **PAIR-PI** (Black-Box). Both explicitly test the vulnerabilities you described, jointly optimizing to force a clean Phase 1 trace while hijacking Phase 2 via the cached context. RedVisor maintains a 0% ASR against both, proving the clean reasoning trace neutralizes the malicious context. Please refer to **our rebuttal to Reviewer fAm1** for detailed settings and results.
>
> We will explicitly incorporate these latency breakdowns, OOD evaluations, and expanded adaptive attacks into the final manuscript.

---

> > ### Author Rebuttal · Reviewer_USTm · 2026-04-03
> >
> > My concerns have been adequately addressed.

---

### Official Review · Reviewer_fAm1 · 2026-03-13

**Soundness:** 3
**Presentation:** 3
**Significance:** 3
**Originality:** 3
**Overall Recommendation:** 4
**Confidence:** 4

**Summary:**

This paper studied how prompt injection defense can be made both effective and efficient without the heavy overhead. This paper proposes RedVisor, a two-phase framework with a lightweight top-layer adapter: 1) generates a reasoning trace that identifies malicious segments, 2) mutes the adapter and reuses the KV cache to produce a safe final response. Experiments on various benchmarks show that RedVisor outperforms other defenses.

**Compliance With Llm Reviewing Policy:**

Affirmed.

**Final Justification:**

The rebuttal addressed my concerns, and I am keeping my original recommendation.

**Key Questions For Authors:**

See weakness.

**Limitations:**

yes

**Strengths And Weaknesses:**

**Strengths:**
This paper studies defense against prompt injection attacks. It proposes a unified framework that synthesizes the explainability of detection systems with the seamless integration of prevention strategies. Results show that, compared to existing methods, the proposed method achieves better detection accuracy and significantly reduces the attack success rate, while maintaining high practicality and inference efficiency.


**Weaknesses:**
The experimental evaluation of adaptive attacks is limited. This work only considers GCG, which is insufficient to demonstrate robustness against a broader range of adaptive attack strategies. Including additional attack methods would strengthen the evaluation.

---

> ### Author Rebuttal · Authors · 2026-03-30
>
> Dear Reviewer fAm1,
>
> We sincerely thank you for your thoughtful review. We appreciate your recognition of RedVisor's core contribution as a unified framework that synthesizes the explainability of detection systems with the seamless integration of prevention strategies. We are also glad you highlighted our method's ability to achieve superior detection accuracy and significantly reduce the Attack Success Rate (ASR), all while maintaining high practicality and strict inference efficiency compared to existing baselines.
>
> We address your valuable feedback regarding the scope of our adaptive attack evaluation below.
>
> **Broader Adaptive Attack Evaluation**
> We appreciate your constructive suggestion to evaluate against a broader range of adaptive attack strategies. In our initial submission, we utilized GCG because it serves as a rigorous, standard white-box optimization baseline for testing worst-case parameter vulnerabilities. However, we agree that demonstrating robustness against more diverse, semantically fluent adaptive attacks further solidifies the strength of our defense.
>
> To comprehensively address this and strengthen the evaluation, we followed recent methodologies in the prompt injection community [1] to expand our adaptive attack suite. Specifically, we adapted two state-of-the-art frameworks, AutoDAN [2] and PAIR [3], for the Prompt Injection (PI) threat model. In these adaptations, the attack objective strictly shifts from safety-bypassing (jailbreaking) to instruction-hijacking, and the generated adversarial payloads consist of fluent, natural language rather than token-level gradient substitutions.
>
> * **AutoDAN-PI (White-Box, Fluent Optimization):** We adapted the AutoDAN framework to optimize the injected data segments. AutoDAN utilizes a Hierarchical Genetic Algorithm to maintain high semantic fluency and stealth, while concurrently optimizing the exact same dual-objective loss designed to break RedVisor (i.e., forcing the Phase 1 adapter to output "No injection detected" while simultaneously forcing the Phase 2 backbone to output the malicious payload).
> * **PAIR-PI (Black-Box, LLM-Driven Semantic Deception):** We adapted the Prompt Automatic Iterative Refinement (PAIR) framework. In this setup, a powerful Attacker LLM iteratively queries the RedVisor pipeline. Based on the system's responses, the Attacker LLM dynamically refines highly contextual and semantic prompt injections, attempting to logically deceive the Phase 1 reasoning inspection.
>
> **Results (Attack Success Rate on Alpaca-Farm Subset):**
> | Backbone LLM | GCG (Baseline) | AutoDAN-PI | PAIR-PI |
> | :-: | :-: | :-: | :-: |
> | Llama3-8B | 0% | 0% | 0% |
> | Mistral-7B | 0% | 0% | 0% |
> | Qwen-2.5-7B | 0% | 0% | 0% |
>
> As demonstrated by the results, RedVisor maintains a robust 0% ASR, completely neutralizing these advanced, semantically optimized adaptive attacks. The explicit Phase 1 reasoning trace consistently identifies and isolates the malicious intent, proving that RedVisor's architecture is structurally resilient against both mathematical gradient manipulation (GCG), genetic optimization (AutoDAN), and logical semantic deception (PAIR). We will prominently include this expanded adaptive attack suite in the revised manuscript to further strengthen the evaluation as you suggested.
>
> **References:**
>
> [1] Adaptive Attacks Break Defenses Against Indirect Prompt Injection Attacks on LLM Agents.
>
> [2] AutoDAN: Generating Stealthy Jailbreak Prompts on Aligned Large Language Models.
>
> [3] Jailbreaking Black Box Large Language Models in Twenty Queries.

---

> > ### Author Rebuttal · Reviewer_fAm1 · 2026-04-03
> >
> > Thank you for the author's reply. My concern has been resolved.

---

### Decision · Program_Chairs · 2026-04-30

**Decision:**

Accept (regular)

**Comment:**

The paper proposes a two-phase prompt injection defense that uses a top-layer gated adapter to generate an explainable reasoning trace and then mutes the adapter to produce the final response conditioned on that trace, enabling zero-copy KV cache reuse. Reviewers agree that the core idea is novel and well-motivated, and the vLLM integration is a solid system-level contribution. The rebuttal has addressed major concerns with additional experiments. One reviewer’s concern about missing direct CoT fine-tuning baseline was partially resolved. Overall, the paper has a clear technical contribution. Therefore, I recommend weak accept.